# Mechanisms of Organophosphate Toxicity and the Role of Acetylcholinesterase Inhibition

**DOI:** 10.3390/toxics11100866

**Published:** 2023-10-18

**Authors:** Vassiliki Aroniadou-Anderjaska, Taiza H. Figueiredo, Marcio de Araujo Furtado, Volodymyr I. Pidoplichko, Maria F. M. Braga

**Affiliations:** 1Department of Anatomy, Physiology, and Genetics, F. Edward Hébert School of Medicine, Uniformed Services University of the Health Sciences, Bethesda, MD 20814, USA; vanderjaska@usuhs.edu (V.A.-A.); volodymyr.pidoplichko.ctr@usuhs.edu (V.I.P.); 2Department of Psychiatry, F. Edward Hébert School of Medicine, Uniformed Services University of the Health Sciences, Bethesda, MD 20814, USA

**Keywords:** organophosphates, nerve agents, acetylcholinesterase, respiratory depression, neurotoxicity, status epilepticus, excitotoxicity, calcium dyshomeostasis, oxidative stress, neuroinflammation

## Abstract

Organophosphorus compounds (OPs) have applications in agriculture (e.g., pesticides), industry (e.g., flame retardants), and chemical warfare (nerve agents). In high doses or chronic exposure, they can be toxic or lethal. The primary mechanism, common among all OPs, that initiates their toxic effects is the inhibition of acetylcholinesterase. In acute OP exposure, the subsequent surge of acetylcholine in cholinergic synapses causes a peripheral cholinergic crisis and status epilepticus (SE), either of which can lead to death. If death is averted without effective seizure control, long-term brain damage ensues. This review describes the mechanisms by which elevated acetylcholine can cause respiratory failure and trigger SE; the role of the amygdala in seizure initiation; the role of M1 muscarinic receptors in the early stages of SE; the neurotoxic pathways activated by SE (excitotoxicity/Ca^++^ overload/oxidative stress, neuroinflammation); and neurotoxic mechanisms linked to low-dose, chronic exposure (Ca^++^ dyshomeostasis/oxidative stress, inflammation), which do not depend on SE and do not necessarily involve acetylcholinesterase inhibition. The evidence so far indicates that brain damage from acute OP exposure is a direct result of SE, while the neurotoxic mechanisms activated by low-dose chronic exposure are independent of SE and may not be associated with acetylcholinesterase inhibition.

## 1. Introduction

Organophosphates are organic compounds that contain phosphorus. While biomolecules that are fundamental to life, such as DNA or ATP, contain phosphate groups, the term “organophosphates” (commonly abbreviated as OPs) refers primarily to human-made organophosphorus compounds that are used as herbicides, pesticides, and insecticides in agriculture and public health programs, in veterinary medicine (e.g., for the control of ticks and fleas), and in the production of flame retardants, plasticizers, lubricants, or fuel additives. Depending on a number of factors—major among them is the specific type of the OP and its concentration—OPs can be toxic to humans, animals, plants, insects, and, in general, the environment. Their toxicity to humans has led to the development of the organophosphorus nerve agents, which are the most lethal chemical warfare weapons [1]. Common among all OPs is their primary mechanism of action, which is the inhibition of acetylcholinesterase (AChE, [EC 3.1.1.7]), the enzyme that catalyzes the hydrolysis of the neurotransmitter acetylcholine (ACh), thus preventing the excessive elevation of ACh in cholinergic synapses (neuron to neuron or neuron to muscle synapses where ACh is released) [1]. In this review, we will describe the mechanisms of OP toxicity—with emphasis on neurotoxicity—that are triggered by AChE inhibition, and briefly discuss other mechanisms that may not depend on AChE inhibition. Understanding these mechanisms is crucial for the development of new and effective medical countermeasures—or for the improvement of existing ones—to mitigate or prevent the toxic effects of OPs.

## 2. Cholinergic Mechanisms of OP Toxicity

The main target of OPs is AChE, the enzyme that breaks down ACh into acetic acid and choline, terminating synaptic signal transmission mediated by ACh in neuromuscular junctions, in the autonomic (mainly parasympathetic) nervous system, and in the brain. The OP binds to and phosphorylates a nucleophilic serine at the catalytic site of the enzyme [2], thus preventing the hydrolysis of ACh and resulting in excessive elevation of ACh in cholinergic synapses. Insects, whose central nervous system utilizes ACh as the major excitatory neurotransmitter [3,4,5], are killed instantly by OPs, primarily due to hyperstimulation of nicotinic cholinergic receptors, which are the most abundant in their central nervous system [3,4,5]. In mammals, a cascade of life-threatening effects follows acute exposure to an OP—involving hyperstimulation of both nicotinic and muscarinic cholinergic receptors—which we will discuss below.

### 2.1. Depression of Respiration

As ACh plays a central role in every aspect of respiratory control [6], the most life-threatening effect of the acute increase in ACh is the depression of respiration, which results from cholinergic hyperstimulatory effects on both the central and the peripheral nervous system and can lead to respiratory failure. Respiration is controlled by the respiratory center in the brainstem, which monitors the levels of oxygen, carbon dioxide, and pH in the blood and makes adjustments to the breathing rate and depth to maintain homeostasis and meet the body’s metabolic demands. The precise mechanism by which acute OP poisoning depresses the activity of the respiratory center is not known. The preBötzinger complex, a cluster of neurons in the lower part of the brainstem (the ventrolateral region of the medulla oblongata), which is considered to be the respiratory center pacemaker (it generates the inspiratory rhythm), has both muscarinic and nicotinic receptors [7,8,9,10], via which excess ACh can alter its activity, leading to central apnea [6,11]. Considering that activation of nicotinic receptors in the central respiratory center has mostly excitatory effects (increases respiratory frequency [12]) and that these receptors desensitize [13,14,15], a simplified possibility could be that excess ACh initially causes rapid breathing via nicotinic receptor hyperstimulation, followed by nicotinic receptor desensitization and decreased respiratory drive. At the same time, overactivation of muscarinic receptors appears to play a key role in central apnea [16,17,18]; injection of the nerve agent sarin into the preBötzinger region of rabbits caused respiratory arrest, which was reversed by atropine, a muscarinic receptor antagonist [16]. In addition, atropine, which can cross the blood–brain barrier, but not methylatropine, which antagonizes muscarinic receptors only peripherally, reversed the decrease in respiratory rate induced by low-dose OP exposure [17]; this [17] and other [18] studies demonstrate the key role that the central respiratory center plays in the OP-induced depression of respiration and the importance of muscarinic receptors in this effect. An increased synaptic inhibitory output in the preBötzinger complex due to muscarinic receptor-mediated strengthening of synaptic excitation of glycinergic inhibitory neurons may play a role in the process by which muscarinic receptor hyperstimulation can lead to respiratory arrest [10].

Although death due to respiratory failure after OP poisoning appears to be a centrally mediated process, the peripheral effects of accumulated ACh (known as peripheral cholinergic crisis) also contribute to respiratory failure [1]. Thus, hyperstimulation of muscarinic receptors by ACh released by neurons of the parasympathetic system, in addition to producing excessive salivation, lacrimation, sweating, rhinorrhea, bradycardia, and hypotension, also causes bronchorrhea, bronchoconstriction, and alveolar edema [19,20,21], leading to increased airway resistance and reduced airflow. Hyperstimulation of nicotinic receptors at the junctions of spinal nerves (whose activity is driven by the respiratory center) with respiratory muscles (diaphragm and intercostal muscles) causes repeated sustained contractions followed by nicotinic receptor desensitization, which can lead to muscle weakness and flaccid paralysis [11]. Lastly, the severely compromised respiration may be further compounded by intense seizures (see next section), as clinical studies have found that seizures in limbic structures (not caused by OP exposure) can inhibit breathing and lead to central apnea [22,23].

### 2.2. Induction of Seizures and Status Epilepticus

In addition to the peripheral effects and the disruption in the function of the central respiratory center described above, the excessive elevation of ACh in cholinergic synapses in the brain can induce seizures and severe status epilepticus (SE) [1]. Certain brain areas may play a more significant role than others in initiating seizure activity after OP exposure and propagating it to other regions of the brain, according to their seizurogenic propensity and centrifugal connections to other brain areas, the degree of their excitatory cholinergic innervation, and other factors. The amygdala, a limbic structure known for its pivotal role in emotional regulation [24], has a high propensity for generation and propagation of seizures [25]. Different lines of evidence suggest that the amygdala is a key region in seizure initiation after OP exposure [26,27,28,29]. McDonough et al. (1987) demonstrated that convulsions could be elicited by microinjection of nerve agents into the basolateral nucleus of the rat amygdala but not by similar microinjections into the piriform cortex, hippocampus, or other brain regions [26]. Measurements of AChE activity after exposure to the nerve agent soman showed that AChE inhibition in the basolateral amygdala is essential for induction of SE; rats that were injected with soman but did not develop SE showed no significant inhibition of AChE in the basolateral amygdala, while in the hippocampus and the piriform cortex, AChE activity had diminished to levels comparable to those seen in rats that developed SE [27] (Figure 1A). Another piece of evidence pointing to the importance of the amygdala in seizure initiation after OP exposure is that the amygdala displays a rapid increase in extracellular glutamate (the neurotransmitter that reinforces and sustains seizures; see next paragraph) after exposure to soman, and this increase occurs earlier in comparison with that in the hippocampus [28]. Lastly, in simultaneous recordings of field potentials from the rat basolateral amygdala and the CA1 hippocampal area, in brain slices in vitro, we observed that application of soman caused prolonged, seizure-like rhythmic discharges in the amygdala but only interictal-like spiking in the hippocampus [29] (Figure 2). In addition to the amygdala, other brain regions appear to be important in the initiation of seizures after OP exposure, such as the area tempestas or the perirhinal and piriform cortices [30,31].

The mechanism by which excess ACh in the brain after OP exposure leads to SE (OP-SE) is not quite clear. Seizures are generated when the balance between excitatory and inhibitory activity in the brain is disrupted. The major excitatory neurotransmitter in the brain is glutamate, while the major inhibitory neurotransmitter is GABA. How does excessive elevation of ACh lead to excessive increase in glutamate-mediated excitatory activity and/or reduction in GABA-mediated inhibition? There are both nicotinic and muscarinic receptors in the brain, located postsynaptically or presynaptically on glutamatergic and GABAergic neurons [32,33,34,35,36]. However, it is primarily muscarinic receptors that mediate the initiation of seizures after OP exposure; muscarinic but not nicotinic receptor antagonists prevent seizure induction if administered before OP exposure and can halt seizure activity if administered shortly after the onset of SE [37,38]. If a muscarinic antagonist is offered with some delay (beyond 10~20 min after SE onset), its seizure-suppressing effectiveness decreases, or the treatment is completely ineffective [39]; this is because, by that time, glutamatergic hyperexcitation has become well established, reinforcing and sustaining itself and SE [40].

Glutamatergic hyperactivity can be induced in a number of ways, such as by excessive depolarization of neurons releasing glutamate, indirect depolarization of such glutamatergic neurons secondarily to reduction in GABA-mediated inhibitory transmission (e.g., by presynaptic inhibition of GABA release), or by presynaptic inhibition of glutamate release. By what mechanisms does muscarinic receptor hyperstimulation after acute OP exposure produce any of these effects, and what subtypes of muscarinic receptors are involved? There are five subtypes of muscarinic receptors (M1 to M5 [33,35]). These receptors are not ionotropic (their structure does not consist of a channel through which ions can flow producing depolarization or hyperpolarization) but, instead, they are coupled to G proteins via which they can affect ion channels, altering neuronal excitability and/or presynaptic neurotransmitter release [33,35]. There is evidence suggesting that the M1 subtype plays an important role in the induction of glutamatergic hyperactivity by OPs [41]. In brain slices of the amygdala, in vitro, bath application of paraoxon initially caused a transient increase in spontaneous GABAergic activity recorded from principal/pyramidal neurons in the basolateral nucleus, immediately followed by a lasting increase in spontaneous glutamatergic activity, resulting in an increased ratio of the charge transferred by spontaneous EPSCs over the charge transferred by spontaneous IPSCs [41]. These effects were blocked in the presence of atropine, a non-selective muscarinic receptor antagonist, or in the presence of VU0255035 [41] (Figure 3), which is a highly selective M1 receptor antagonist [42]. The increase in spontaneous glutamatergic transmission by paraoxon could be the result of M1 receptor–mediated depolarization of glutamatergic neurons via enhancement of the mixed Na^+^/K^+^current, Ih; potentiation of the Ih current by M1 receptor activation has been shown in pyramidal neurons of the basolateral amygdala [41] and the CA3 hippocampal area [43]. Concomitantly, excessive activation of M1 muscarinic receptors may cause presynaptic facilitation of glutamate release via inhibition of a potassium channel, the M-current) (which will increase presynaptic Ca^++^ influx through voltage-gated Ca^++^ channels), as it has been shown by application of paraoxon to hippocampal slices [44].

The selective M1 antagonist, VU0255035, has also been tested in vivo [41,42]. Pretreatment of rats with VU0255035 significantly suppressed behavioral seizures induced by either soman or paraoxon for at least 40 min after exposure, delaying the development of SE [41]. Similar results have been reported in mice treated with VU0255035 before administration of the muscarinic agonist pilocarpine as the convulsant [42]. Despite pretreatment with VU0255035, low-intensity seizures occurred after convulsant administration in both studies [41,42], and SE manifested itself but with a significant delay. These results suggest that activation of muscarinic receptors other than the M1 subtype may initiate a low-level increase in glutamatergic activity, which gradually reinforces itself, eventually developing into SE. It is also possible, however, that higher doses of VU0255035 would completely prevent the occurrence of SE. The studies by Sheffler et al. (2009) [42] and Miller et al. (2017) [41] employed doses that did not produce cognitive side effects in mice [42]. This is important if antimuscarinic compounds are to be used as a pretreatment in individuals who are at risk of OP exposure. Whether or not higher doses of VU0255035 can completely prevent OP-SE (given as a pretreatment) or block OP-SE if administered soon after its onset—which would answer whether the M1 receptors alone mediate the initiation of seizures—remains to be determined. Nevertheless, it is important that VU0255035, at a dose that does not appear to produce side effects, can delay the development of OP-SE, allowing time for medical intervention.

### 2.3. Seizures and Status Epilepticus as the Primary Mediators of OP Neurotoxicity

There is ample evidence to indicate that SE is the main cause of neurotoxicity after acute OP exposure, which can lead to long-term brain damage [45] that may be permanent and even worsen over time [46,47]. From a group of rats exposed to soman, a small percentage did not develop SE, despite the appearance of toxic signs and a significant reduction in AChE activity in the piriform cortex and the hippocampus; seven days after the exposure, the rats who developed SE had severe neuronal degeneration in a number of brain regions and neuronal loss in the basolateral amygdala and the CA1 hippocampal area, while the soman-injected rats that did not develop SE had no neurodegeneration anywhere in the brain or neuronal loss in the amygdala and hippocampus [27] (Figure 1B1–B3). Similarly, de Araujo Furtado et al. (2010) found that most of the rats that developed SE after soman exposure displayed axonal degeneration and continued to develop epilepsy (significant presence of spontaneous recurrent seizures), while none of the rats that did not develop SE exhibited these pathologies [48]. Furthermore, in OP-exposed rats that develop SE, brain damage can be prevented only if SE is terminated by anticonvulsant administration shorty (1 to 5 min) after the onset of seizures [49,50] or by delayed anticonvulsant treatment (up to ~1 h after SE onset) that not only stops SE but is also successful in preventing or minimizing its reoccurrence [47,51,52]. Consistent with these studies which point to the pivotal role of SE in brain damage by acute OP exposure, Shih et al. (2003) found that there was a strong association between brain damage and seizure control after nerve agent exposure; animals had higher incidence of—and more severe neuropathology—when the seizures were not successfully controlled [53].

The main mechanism by which SE (of any etiology) causes brain damage is excitotoxity [54,55], the pathological process via which neurons die due to excessive stimulation by excitatory neurotransmitters, primarily glutamate. During intense seizure activity, overactivation of AMPA, kainate, and NMDA receptors by glutamate results in a massive Ca^++^ influx into neurons, via subtypes of AMPA and kainate receptors that are permeable to Ca^++^ and, mainly, via NMDA receptor channels [56,57]. Calcium overload initiates a cascade of events that lead to cell damage and death via both necrotic and apoptotic pathways. Necrosis is a rapid cell death involving a largely uncontrolled cell damage, as excess Ca^++^ disrupts the function of mitochondria, leading to ATP depletion, overproduction, and inefficient elimination of reactive oxygen species and oxidative stress (with consequences such as damage of cellular proteins, lipids and DNA; for a review on DNA damage by OPs, see [58]); neurons swell and eventually burst, releasing inflammatory contents into the extracellular space and surrounding tissue [57,59]. Apoptotic cell death is a more controlled and “programmed” process [59,60]. In the context of SE, the major events in the apoptotic pathway triggered by excess neuronal Ca^++^ are mitochondrial dysfunction, oxidative stress, and cytochrome c release, leading to activation of caspases. Caspases cleave intracellular organelles, proteins, and DNA, which are then packaged and cleared by phagocytosis. Excess Ca^++^ also causes endoplasmic reticulum stress; activates calpains, which can degrade vital cellular components; and leads to activation of the death-receptor pathway. In contrast to necrosis, where cell contents spill out and provoke inflammation, apoptosis is a more contained process that does not cause inflammation, as there is no spilling of proinflammatory components into the extracellular space. For reviews of the detailed mechanisms involved in the different pathways that lead to neuronal death by SE, the molecular effectors and signaling routes that these pathways share, and the blurring lines between traditional definitions of these pathways, see references [57,59,60,61,62]. It should also be noted that the relative contribution of the different pathways to neuronal death may be affected by the age of the organism and the maturation stage of the neuron [63,64,65].

In addition to neuronal Ca^++^ overload and its consequences, another mechanism that can contribute to neurotoxicity by SE involves the role of glia cells. During and after SE there is a sustained increase in intracellular Ca^++^ in astrocytes, which enhances astrocytic glutamate release, contributing to excitotoxic damage of neurons, probably by stimulation of neuronal extrasynaptic GluN2B-containing NMDA receptors [66]; since astrocytes have Ca^++^-permeable AMPA receptors, NMDA receptors, and metabotropic glutamate receptors which stimulate Ca^++^ release from intracellular stores [67], the main cause of calcium increase in astrocytes during SE could be hyperstimulation of these receptors by excess extracellular glutamate. Astrocytes also play a crucial role in clearing up excess glutamate from the synaptic cleft and the extracellular space via glutamate transporters [68]; this process is impaired during SE, contributing to excitotoxicity [69,70]. Microglia, the resident immune cells in the central nervous system, are also activated during SE, and, along with astrocytes, they are the primary players in neuroinflammation, which plays an important role in neuronal death [69,71,72,73].

The complex interactions among neurons, microglia, astrocytes, and other cells involved in the inflammatory immune response during SE, as well as the interactions among the various inflammatory molecules that these cells release, are not completely understood. During SE, microglia and astrocytes are activated by excess glutamate acting on its receptors of glia cells, or by proinflammatory cytokines and chemokines (proteins that can travel in body fluids and bind to specific receptors on target cells [74]), which are released from neurons that are under oxidative stress. Activated microglia and astrocytes then produce more reactive oxygen species, leading to a vicious cycle of oxidative stress and progression of the inflammatory response, as these glia cells also release proinflammatory cytokines and chemokines [71,72]. Cytokines can exacerbate seizures and neurotoxicity by a number of mechanisms including enhancement of Ca^++^ influx through GluN2B-containing NMDA receptors [75]. Another component of the inflammatory response is the generation of reactive nitrogen species by neurons and glia that are under stress [76,77]. At the same time that these inflammatory mediators are produced, the breakdown of the blood–brain barrier allows peripheral immune cells to enter the brain, promoting inflammation [78]. Hypoxia also promotes neuroinflammation [79,80]; the severe hypoxia during OP-SE, due to both unrelenting seizures and the peripheral cholinergic crisis, exacerbates oxidative stress and neuroinflammation. Acute OP intoxication can also cause systemic inflammation that can result in myocarditis, pericarditis, pulmonary edema, or acute pancreatitis [71], which could potentially exacerbate neuroinflammation due to systemic release of proinflammatory cytokines. Inflammation contributes significantly to neuronal death by various mechanisms [81]. For example, cytokines and chemokines can directly induce apoptosis [62,82,83]; activated microglia release reactive oxygen species [84], which induce oxidative stress in neighboring neurons leading to neuronal death; activated microglia may phagocytize stressed but still viable neurons [85,86]; and certain components of the complement system (part of the innate immune system) that is activated during inflammation can cause direct lysis of neurons or draw immune cells like macrophages to the site of injury, which damage and eliminate neurons through phagocytosis [87,88,89].

Lastly, cerebral blood flow alterations associated with SE contribute to neurotoxicity. For example, after 2 h of SE induced by pilocarpine, the infragranular layers of the rat somatosensory cortex displayed reduced vascular perfusion indicative of ischemia, which was accompanied by rapid necrotic neuropathology; the supragranular layers displayed increased vascular perfusion and a more delayed neurodegeneration with apoptotic features [90]. In SE from acute OP exposure, neurotoxicity due to ischemia and hypoxia may play a more significant role compared with SE of other etiologies because of the severe respiratory depression by central and peripheral mechanisms, in addition to seizures, which exacerbate hypoxic conditions [91].

## 3. Mechanisms of OP Toxicity That Do Not Depend on Status Epilepticus or AChE Inhibition

Although this review has focused on OP toxic mechanisms that depend on AChE inhibition and the central role of SE in neurotoxicity, it is important to note that excess ACh alone (without SE) or other mechanisms that are independent of AChE inhibition may contribute to toxicity and neurotoxicity. Depending on a number of factors, such as the type of the OP, the degree of AChE inhibition, or even individual variability in the rate of new AChE synthesis, restoring normal levels of AChE activity after acute OP exposure may take weeks or months in humans (it takes 1 to 2 weeks for AChE activity to recover after acute exposure of rats to soman [92], which corresponds to almost one human year [93,94]). It is conceivable, then, that sustained elevated ACh above normal levels could directly harm neurons by increasing intracellular Ca^++^ through activation of nicotinic and muscarinic receptors, disrupting the balance of other neurotransmitter systems, and other possible mechanisms. Prolonged stimulation of nicotinic receptors by elevated ACh, resulting in downregulation of the receptors, may be responsible for the muscle weakness—importantly including the muscles of respiration—that occurs in the intermediate syndrome [95]. However, elevated ACh may also have beneficial effects, such as the anti-inflammatory effects it exerts via activation of α7 subunit-containing nicotinic receptors on macrophages [71,72,96]. It is, therefore, unclear whether beneficial or detrimental effects of excess ACh predominate in the absence of SE.

In regard to neurotoxic effects of OPs that are independent of both SE and AChE inhibition, insights may be gained from studies of low-dose, long-term exposure, which does not involve seizures. As in acute OP exposure, oxidative stress and inflammation play an important role in neuronal damage caused by chronic OP exposure [71,97,98,99]; oxidative stress in chronic exposure may be at least partly dependent on AChE inhibition, since in OP-formulating pesticide workers, it is associated with reduced erythrocyte AChE activity [100]. However, others have found no significant association between neurobehavioral deficits caused by occupational exposure to OPs and reduced blood cholinesterase activity [101]. Investigations of low-dose, chronic exposure to OPs in animal models also show long-term toxic effects [71,102]. The mechanisms that produce these effects are not completely clear. For example, Phillips et al. (2019) found that repeated low-dose exposure of rats to diisopropyl fluorophosphate caused sustained elevations of Ca^++^ in hippocampal neurons, which were accompanied by symptoms of depression and cognitive deficits [103]. The increase in intracellular Ca^++^ was partly due to influx through NMDA receptors and to a greater extent due to Ca^++^ release from intracellular stores [103]. It is not known whether elevated acetylcholine itself and/or an increased basal glutamatergic excitation contributed to these effects. Direct mechanisms through which OPs may contribute to neurotoxicity are better revealed in reduced preparations, where AChE inhibition can have little or no effect. For example, in cell cultures of primary fetal human astrocytes, OP insecticides increased expression of inflammatory proteins [104], which is unlikely to be via cholinergic mechanisms. It should also be noted that although all OPs share the common toxicity mechanism of AChE inhibition, there are some differences between them in what macromolecules they target and to what extent; these differences could somewhat modify the cascade of events after AChE inhibition or differentially engage non-cholinergic mechanisms [105].

## 4. Conclusions

In this review, we have provided an overview of the major mechanisms of OP toxicity. Acute exposure to OPs can lead to death due to respiratory depression and/or due to SE, both of which are caused by the inhibition of AChE and the excessive elevation of ACh. If death is prevented but not with concomitant effective control of seizures, brain damage will ensue, manifested as neuronal loss, restructuring of neuronal circuits, hyperexcitability, epileptogenesis, and behavioral deficits. Some of the mechanisms of SE-induced neurotoxicity after acute OP exposure (e.g., oxidative stress and inflammation) are common with the mechanisms that low-dose chronic exposure causes neuropathology, without involving SE. However, this does not mean that SE-independent mechanisms (mechanisms that are not triggered by SE) play a significant role in causing damage after acute OP exposure; the evidence that supports this view is that there is no brain damage if acute exposure does not induce SE or if SE is treated effectively. In low-dose repeated exposures, SE-independent neurotoxicity mechanisms can cause brain damage probably because they are activated chronically; in addition, long-term systemic toxicity could contribute to neuropathology.

In acute OP exposure, effective control of SE that will prevent brain damage and long-term morbidity can be achieved by administration of a benzodiazepine shortly after the onset of SE [49,50,106]. If medical treatment of SE is somewhat delayed—as may unavoidably be the case in the event of a mass exposure—antiglutamatergic therapies are more effective in controlling seizures and preventing brain damage [106,107]; for this reason, antiglutamatergic therapies should be given serious consideration for use as first-line delayed treatment. If the anticonvulsant treatment—because of its timing or the receptors and mechanisms it targets—does not prevent or halt the neurotoxic cascades elicited by SE (biomarkers are probably indispensable in this determination [108]), then anti-inflammatory therapies [109] will be essential to limit the damage.

## Figures and Tables

**Figure 1 toxics-11-00866-f001:**
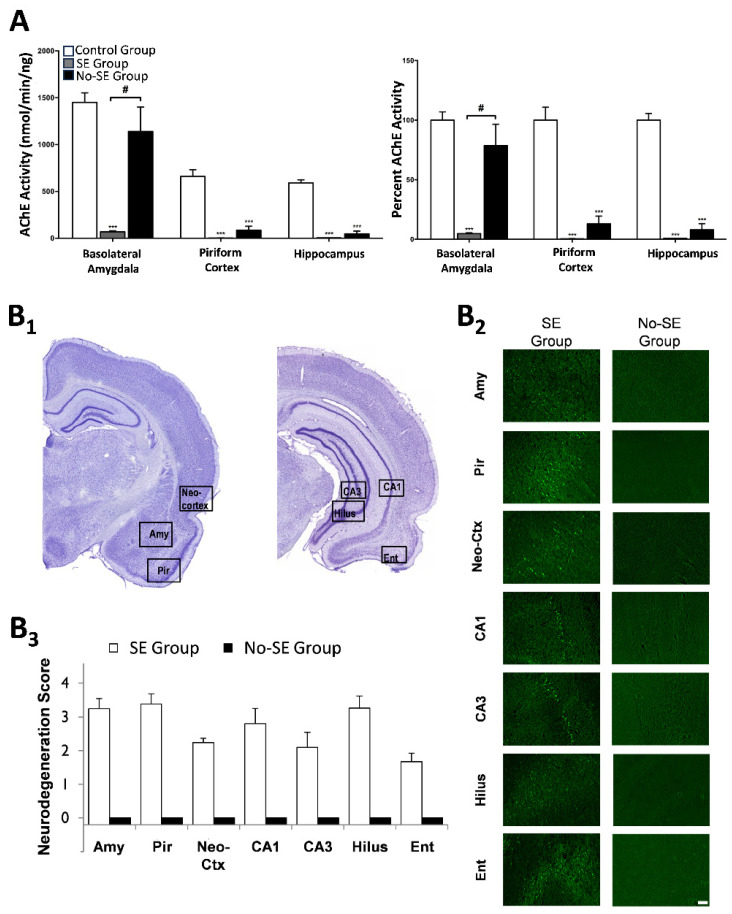
Rats who are acutely exposed to soman but do not develop status epilepticus (SE) have no significant reduction in AChE activity in the basolateral amygdala and do not display any neurodegeneration. (**A**) The left panel shows the raw data of AChE activity, and the right panel shows the percent change compared to the control group (not exposed to soman). For the SE group, measurements were made at the onset of stage 3 seizures (10~20 min after soman exposure); these rats had significantly reduced AChE activity in all three brain regions examined compared with controls. For the no-SE group (soman-exposed rats that did not develop SE), measurements were made 30 min after soman exposure; these rats had significantly reduced AChE activity in the piriform cortex and the hippocampus but not in the BLA. *** *p* < 0.001 compared to the control group; # *p* < 0.05 for the difference between the SE group and the no-SE group. The B panels show neuronal degeneration in the SE group and the no-SE group, 7 days after soman exposure. (**B1**) Panoramic photomicrographs of Nissl-stained sections showing the brain regions where neurodegeneration was evaluated. (**B2**) Representative photomicrographs of Fluoro-Jade C-stained sections for the two soman-exposed groups. Total magnification is 100×, and scale bar is 50 mm. (**B3**) Neurodegeneration scores for the amygdala (Amy), piriform cortex (Pir), entorhinal cortex (Ent), CA1 and CA3 hippocampal subfields, hilus, and neocortex (neo-Ctx) in the SE group and the no-SE group (adapted from Prager et al., 2013 [27]).

**Figure 2 toxics-11-00866-f002:**
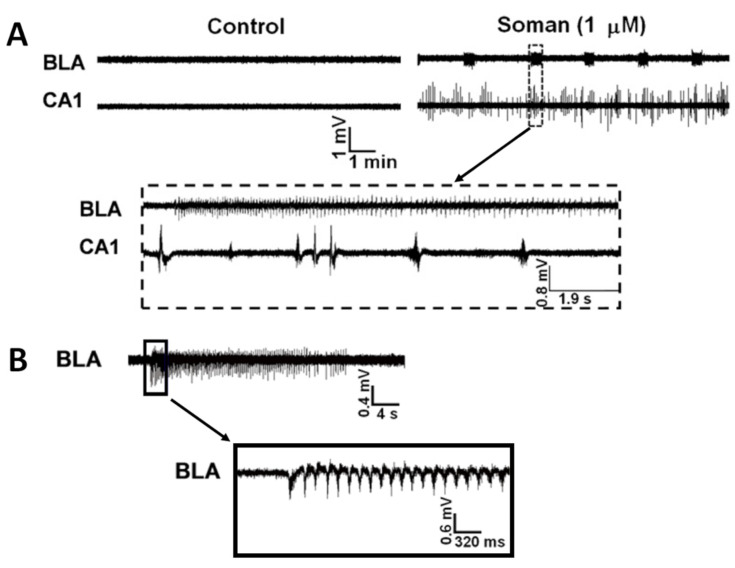
Soman induces ictal-like activity in the amygdala and interictal-like activity in the hippocampus. Extracellular field recordings, in gap-free mode, were simultaneously obtained in the basolateral amygdala (BLA) and the stratum pyramidale of the CA1 hippocampal area in slices containing both regions. Exposure to 1 mM soman for 30 min induced spontaneous, prolonged episodes of synchronous neuronal discharges resembling seizures in the BLA. In the CA1 area, soman exposure produced spontaneous, interictal-like bursts. The effects of soman were not reversible; epileptiform activity was maintained after washing out soman and throughout the recording period. (**A**,**B**) are recordings from two different slices. In (**B**), the seizure recorded from the BLA is shown in a more expanded time scale. Note that the substantially larger amplitude of the field potentials recorded from the CA1 area compared to those recorded from the BLA is due to the anatomical organization of the hippocampal circuitry which favors the generation of strong extracellular current dipoles (adapted from Apland et al., 2009 [29]).

**Figure 3 toxics-11-00866-f003:**
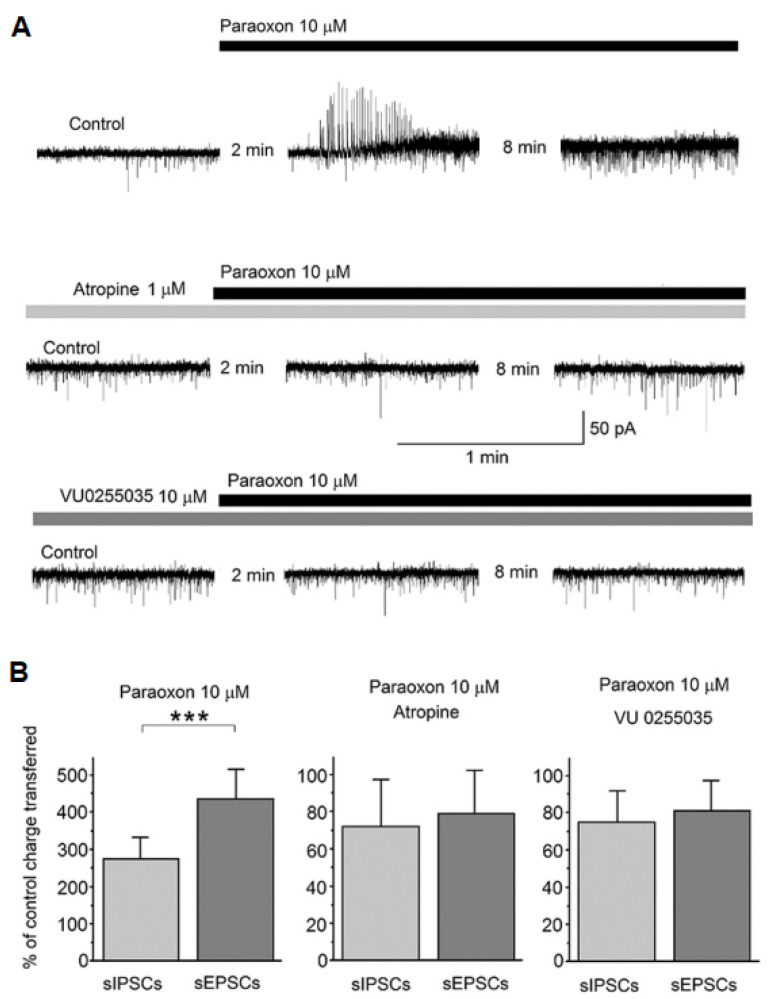
Paraoxon transiently increased spontaneous IPSCs (sIPSCs) in the rat basolateral amygdala (BLA), followed by a lasting increase in spontaneous EPSCs (sEPSCs); both effects were blocked in the presence of atropine or a specific M1 muscarinic receptor antagonist. Whole-cell simultaneous recordings of sIPSCs and sEPSCs were obtained from BLA principal cells (V_h_ = −58). (**A**) Representative examples of the effects of paraoxon on sIPSCs and sEPSCs in control conditions (top trace) and the presence of atropine or VU0255035. Outward currents (upward deflections) are GABAergic and inward currents (downward deflections) are glutamatergic. The number of minutes between the first two traces in each of the three rows is the time lapse from the time point paraoxon was applied, and between the second and third traces it is the interval between the two successive recordings. (**B**) The bar graph on the left shows the effect of 10 μM paraoxon on the total charge transferred by sIPSCs and sEPSCs during a 20 s time window, at 10 min after paraoxon application, expressed as the percentage of the total charge transferred in control conditions (absence of paraoxon). The same is shown in the middle and right bar graphs, but in the presence of 1 μM atropine or 10 μM VU0255035, respectively. The total charge transferred by sIPSCs and sEPSCs was increased by paraoxon (disproportionately, favoring the increase in sEPSCs) only when there was no atropine or VU0255035 in the slice medium. *** *p* < 0.001 for the difference between the increase in sIPSCs and the increase in sEPSCs by paraoxon (adapted from Miller et al., 2017 [41]).

## Data Availability

No new data were created for this review paper.

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
