# Peer review of "Mechanisms of Organophosphate Toxicity and the Role of Acetylcholinesterase Inhibition"

_toxics, 2023, doi:10.3390/toxics11100866_

Round 1

Reviewer 1 Report

The review by Aroniadou-Anderjaska et al bring a nice overview of the current knowledge how OPs influence neurotransmission. I have only several minor comments to be addressed

1.       I recommend adding references after specific general statements and at the beginning of the specific descriptions and not only later in the text. It would be clearer for readers and would ease further reading/literature search.

I will only state some examples, to get the idea where references should be added. For example:

-          line 33-39 addressing the use of OPs “….are used as herbicides, pesticides….” add a reference

-          line 43-47 addressing the AChE inhibition “Common among all OPs…..” add reference; (also in this sentence when first mentioning AChE add its EC nomenclature number in the brackets)

-          line 68-70 addressing “Respiration…” add a reference

-          line 93-95 “….peripheral effects of accumulated Ach also contribute to respiratory failure” add a reference

-          line 99-103 “Hyperstimulation of nicotinic receptors ….causes repeated contractions…”  add a reference

-          line 109-110 “….excessive elevation of ACh …can induce SE” add a reference

-          line 168-170 addressing seizures - add a reference; also for GABA release in line182-186

-          line 192-194 “There is evidence…” - add a reference

-          line 227 “The selective M1 antagonist….has also been tested in vivo” - add a reference

etc.

2.       there is no referring to the Fig 1B1 and B2 in the text (only 1A)? So please add where it is needed

3.       please check the copyright for using figures previously published in other journals/papers before. What does it mean "adapted from", how are they different?

4.       Fig 1B2 – quite low visibility of this part. Perhaps the problem is only in the version for the revision but please check.

5.       Fig 3 B – it is not that clear why the x-axes are in different values for the treatment with POX only (300) and for the treatment of POX + atropine/VU (70)? If it is just blocking effect of cca 4 times after the treatment perhaps to put all in one graph to make this blocking effect clear for readers and visible right away

Reviewer 2 Report

For section 2.1. The authors describe the impact of OP poisoning on brainstem respiratory control.  Is there any evidence that seizure activity propagating to brainstem respiratory control centers, or the amygdala for that matter, affect breathing following OP exposure?

minor comment - an end parenthesis is missing in line 328, page 10
